# Suicide attempt and death by suicide among parents of young individuals with cancer: A population-based study in Denmark and Sweden

**Qianwei Liu**[1,2]*, **Krisztina D. László**[3,4], **Dang Wei**[2], **Fen Yang**[3], **Katja Fall**[2,5], **Unnur Valdimarsdóttir**[2,6,7], **Maria Feychting**[2], **Jiong Li**[8], **Fang Fang**[2]

1 Department of Hematology, Nanfang Hospital, Southern Medical University, Guangzhou, China, 2 Institute of Environmental Medicine, Karolinska Institutet, Stockholm, Sweden, 3 Department of Global Public Health, Karolinska Institutet, Stockholm, Sweden, 4 The Department of Public Health and Caring Sciences, Uppsala University, Uppsala, Sweden, 5 Clinical Epidemiology and Biostatistics, School of Medical Sciences, Örebro University, Örebro, Sweden, 6 Center of Public Health Sciences, School of Health Sciences, Faculty of Medicine, University of Iceland, Reykjavík, Iceland, 7 Department of Epidemiology, Harvard T.H. Chan School of Public Health, Boston, Massachusetts, United States of America, 8 Department of Clinical Medicine-Department of Clinical Epidemiology, Aarhus University, Aarhus, Denmark

* qianweiliu@smu.edu.cn

## Abstract

### Background

The psychological toll on parents of a child receiving a cancer diagnosis is known to be high, but there is a knowledge gap regarding suicidal behavior among these parents. The aim of this study was to investigate the risk of suicide attempt and death by suicide in relation to having a child with cancer.

### Methods and findings

We performed a binational population-based and sibling-controlled cohort study, including all parents with a child diagnosed with cancer in Denmark (1978 to 2016) or Sweden (1973 to 2014), 10 matched unexposed parents per exposed parent (population comparison), and unaffected full siblings of the exposed parents (sibling comparison). Suicide attempt was identified through the Patient Register and the Psychiatric Central Register in Denmark and the Patient Register in Sweden, whereas death by suicide was identified through the Danish Causes of Death Register and the Swedish Causes of Death Register. In population comparison, we used Cox regression to estimate hazard ratios (HRs) and 95% confidence intervals (CIs) of suicide attempt and death by suicide associated with cancer diagnosis of a child, adjusting for sex, age, country of residence, calendar year, marital status, highest attained educational level, household income, history of cancer, history of psychiatric disorder, and family history of psychiatric disorder. The sibling comparison was performed to assess the role of familial confounding in the studied associations. The population comparison consisted of 106,005 exposed parents and 1,060,050 matched unexposed parents, with a median age of 56 at cohort entry and 46.9% male. During the median follow-up of 7.3

**Data Availability Statement:** The data underlying this article were provided by the Statistics Denmark (https://www.dst.dk/en/), Swedish National Board of Health and Welfare (https://www.socialstyrelsen.se/en), as well as Statistics Sweden (http://www.scb.se/). The data cannot be shared publicly due to

the regulations of the above data holder authorities and ethical reasons. Similar data might be requested from these holder authorities for research purposes by researchers who fulfil specific requirements.

**Funding:** This study was supported by the Swedish Cancer Society (grant number: 20 0846 PjF to FF), Karolinska Institutet (Senior Researcher Award and Strategic Research Area in Epidemiology to FF), the China Scholarship Council (grant number: 201700260291 to QL; grant number: 201700260276 to DW), the Novo Nordisk Foundation (grant number: NNF18OC0052029 to JL), the Independent Research Fund Denmark (grant numbers: DFF-6110-00019B, DFF-9039-00010B, and 1030-00012B to JL), the Nordic Cancer Union (grant number: R275-A15770 and R278-A15877 to JL), the Karen Elise Jensens Fond (2016 to JL), and the Swedish Research Council for Health, Working Life and Welfare (grant numbers: 2017-00531 to FF and 2015-00837 to KDL). The funders had no role in study design, data collection and analysis, decision to publish, or preparation of the manuscript.

**Competing interests:** The authors have declared no conflicts of interest.

**Abbreviations:** BMI, body mass index; CI, confidence interval; CNS, central nervous system; HR, hazard ratio; ICD, International Classification of Diseases; IR, incidence rate; MBR, Medical Birth Register; PPV, positive predictive value.

and 7.2 years, we observed 613 (incidence rate [IR], 58.8 per 100,000 person-years) and 5,888 (IR, 57.1 per 100,000 person-years) cases of first-onset suicide attempt among the exposed and unexposed parents, respectively. There was an increased risk of parental suicide attempt during the first years after a child's cancer diagnosis (HR, 1.15; 95% CI, [1.03, 1.28]; $p = 0.01$), particularly when the child was 18 or younger at diagnosis (HR, 1.25; 95% CI, [1.08, 1.46]; $p = 0.004$), when the child was diagnosed with a highly aggressive cancer (HR, 1.60; 95% CI, [1.05, 2.43]; $p = 0.03$), or when the child died due to cancer (HR, 1.63; 95% CI, [1.29, 2.06]; $p < 0.001$). The increased risk did not, however, maintain thereafter (HR, 0.86; 95% CI: [0.75, 0.98]; $p = 0.03$), and there was no altered risk of parental death by suicide any time after the child's cancer diagnosis. Sibling comparison corroborated these findings. The main limitation of the study is the potential residual confounding by factors not shared between full siblings.

## Conclusions

In this study, we observed an increased risk of parental suicide attempt during the first years after a child's cancer diagnosis, especially when the child was diagnosed during childhood, or with an aggressive or fatal form of cancer. There was, however, no altered risk of parental death by suicide at any time after a child's cancer diagnosis. Our findings suggest extended clinical awareness of suicide attempt among parents of children with cancer, especially during the first few years after cancer diagnosis.

## Author summary

### Why was this study done?

- Having a child with a cancer diagnosis is highly stressful for parents.
- Little is known about risk of parental suicidal behavior in relation to cancer diagnosis of a child.

### What did the researchers do and find?

- We performed a binational population-based and sibling-controlled cohort study to investigate the association between cancer diagnosis of a child and the subsequent risk of suicide attempt or death by suicide among the parents.
- We observed an increased risk of parental suicide attempt during the first years after a child's cancer diagnosis, especially when the child was diagnosed during childhood, or with an aggressive or fatal form of cancer, in both the population-based comparison and the sibling comparison.

### What do these findings mean?

- Our findings call for clinical awareness of the risk of suicide attempt among parents of children with cancer, primarily within the first years after cancer diagnosis of the child.

- Future studies are needed to examine the generalizability of our findings to other countries with different healthcare system, sociocultural context, and prevalence of cancer and suicidal behavior as Denmark and Sweden.

## Introduction

A cancer diagnosis of a child is a traumatic and highly stressful event for the parents [1]. Parents of children with cancer may experience physical, mental, and financial challenges [2,3]. These parents, especially mothers, have indeed been reported to have increased psychological distress and risk of psychiatric disorders [1,4–6]. However, although psychological distress and psychiatric disorders are known to increase the risk of suicidal behavior [7,8], only a few studies have so far assessed the association of a child cancer diagnosis with parental risk of suicidal behavior. The existing studies rendered largely conflicting findings, often based on relatively small sample size [5,9,10]. Most of these studies have had limited control for important confounding factors and focused on death by suicide alone, leaving the risk of suicide attempt unraveled [9,10]. Moreover, none of these studies has explored the temporal pattern of suicidal behavior.

Considering the relatively low incidence of suicidal behavior, especially death by suicide, population-based studies with large sample size are needed to provide sufficient statistical power, especially for the analyses of temporal pattern in the risk of suicidal behavior and by cancer characteristics. Cross-validation of findings between populations as well as contrasting findings between suicide attempt and death by suicide in the same population are also of importance. To this end, we took advantage of several national population and health registers in Denmark and Sweden and performed a population-based study to investigate the association between cancer diagnosis of a child and risk of parental suicide attempt and death by suicide, after adjusting for several potential confounders as well as controlling for familial confounding through a sibling comparison. We hypothesize that there is an altered risk of suicidal behavior among parents of children with cancer and that such risk alteration differs by time since cancer diagnosis and according to characteristics of the child's cancer.

## Methods

### Study design and population

In this binational study, we first identified biological parents of all live births during 1973 to 2016 according to the Danish Medical Birth Register (MBR) and during 1973 to 2014 according to the Swedish MBR. Both MBRs were established in 1973 and include almost 99% of births in the countries [11,12]. Information on mothers was available for almost all children in the MBRs. We identified fathers for 98% of the Danish children through the MBR and the Danish Civil Registration System, and for 83% of the Swedish children through linkage to the Swedish Multi-Generation Register. As the parents might have had other children born before the establishment of the MBRs or outside Denmark or Sweden, we identified additional children of these parents through the Danish Civil Registration System and the Swedish Multi-Generation Register. As the Danish Patient Register has nationwide coverage of inpatient care since 1978 (outpatient care since 1995) and the Swedish Patient Register has nationwide coverage of inpatient psychiatric care since 1973 (outpatient specialist care since 2001), we defined

the study period as 1978 to 2016 for Denmark and 1973 to 2014 for Sweden. The prespecified analysis plan of the present study is presented in **S1 Text**.

We first identified all parents that had a child diagnosed with cancer during 1978 to 2016 in Denmark ($N$ = 52,027) or 1973 to 2014 in Sweden ($N$ = 59,428) (**S1 Fig**). If a parent had more than 1 child with a cancer diagnosis, the date of cancer diagnosis of the first child was regarded as the index exposure. As we aimed to study newly diagnosed cancer in a child and the first-onset parental suicide attempt or death by suicide, parents that had a child diagnosed with cancer ($N$ = 283 in Denmark and $N$ = 149 in Sweden) or had a suicide attempt ($N$ = 998 in Denmark and $N$ = 1,169 in Sweden) before the study period were excluded from the main analysis. To ensure relatively complete ascertainment of familial links from the Danish Civil Registration System and Swedish Multi-Generation Register, we also excluded parents born before 1936 in Denmark ($N$ = 1,914) or before 1932 in Sweden ($N$ = 944). Finally, 3 parents in Sweden with conflicting information on their children (i.e., a child died or emigrated before date of birth) were also excluded, leaving 106,005 parents ($N$ = 48,842 in Denmark and $N$ = 57,163 in Sweden) in analysis. We defined the date of child cancer diagnosis as the index date for these exposed parents.

We then identified 2 unexposed comparison groups. For each exposed parent, we randomly selected 10 parents from the study population, who were individually matched to the exposed parent on birth year, sex, and country of residence and had no history of suicide attempt, at least 1 alive child, and no child with cancer, before the index date of the exposed parent (population comparison). As cancer and suicidal behavior might share risk factors including factors shared between family members, we additionally performed a sibling comparison including the exposed parents and their unexposed full siblings. We identified 64,446 unexposed full siblings (of 40,218 exposed parents), who had no history of suicide attempt and at least 1 child alive with no diagnosed cancer, before the index date of their exposed full sibling. We used the index date of the exposed parent as the index date for their matched unexposed parents and unexposed siblings.

We then followed all study participants from the index date, until death, emigration, an incidence of suicide attempt or death by suicide, or the end of follow-up (December 31, 2016 in Denmark and December 31, 2014 in Sweden), whichever came first. Follow-up of the unexposed parents and the unexposed siblings was additionally censored if they had a child diagnosed with cancer during follow-up. These study participants would then contribute person-time to the exposed group from the date of their child's cancer diagnosis.

## Definition of exposure

We used the Danish and Swedish revisions of the International Classification of Diseases (ICD) codes to identify child cancer diagnoses from the Danish and Swedish Cancer Registers [13,14]. As cancers in central nervous system (CNS) and hematological malignancies are the most common types of cancer among young individuals, accounting for around two-thirds of all childhood cancers, for instance, in addition to any cancer, we classified cancers into CNS cancer, hematological malignancy, and other type of cancer. Finally, we also analyzed cancer aggressiveness (low, medium, or high), which was defined according to 5-year survival rate of each cancer type [15]. ICD codes of cancer are listed in **S1 Table**.

## Definition of outcome

Suicide attempt was identified through the Patient Register and the Psychiatric Central Register [16,17] in Denmark and the Patient Register in Sweden [18], whereas death by suicide was identified through the Danish Causes of Death Register and the Swedish Causes of Death

Register, using the Danish and Swedish revisions of the ICD codes (**S2 Table**). To better identify suicide attempt, we additionally included the reason for contact code "4" (deliberate self-harm) in the identification of suicide attempt in Denmark, as suggested [19].

## Covariates

We retrieved information on sex, year of birth, as well as marital status of the study participants from the Danish Civil Registration System and the Swedish Total Population Register. We retrieved information on household income from the Danish Integrated Database for Longitudinal Labor Market Research [20] and the Swedish Register of Incomes and Taxes, and on the highest attained educational level from the Danish Integrated Database for Longitudinal Labor Market Research and the Swedish Education Register. We collected information on history of cancer from the Cancer Registers in Denmark and Sweden and on history of psychiatric disorder and family history of psychiatric disorder from the Danish Patient Register, the Danish Psychiatric Central Register, and the Swedish Patient Register. ICD codes of psychiatric disorders are listed in **S2 Table**. Data on smoking (available since 1991 in Denmark and since 1982 in Sweden) and body mass index (BMI) (available since 2003 in Denmark and during 1982 to 1989 and since 1992 in Sweden) during early pregnancy were retrieved from the MBRs (available only for mothers).

In order to address concerns related to missing data on some covariates, we used multiple imputation with fully conditional specification to produce 5 imputed datasets [21]. We then combined estimates from the imputed datasets using Rubin's rules [22]. The detailed methods of multiple imputation are described in **S2 Text**.

## Statistical analysis

We calculated the incidence rates (IRs) of suicide attempt and death by suicide among the exposed and unexposed parents, through dividing number of cases by accumulated person-years at risk. We visualized the time-varying associations of child cancer diagnosis with the risk of parental suicide attempt and death by suicide using flexible parametric survival models, in both the population and sibling comparisons. In all flexible parametric survival models, a spline with 5 degrees of freedom (4 intermediate knots and 2 knots at each boundary, placed according to quintile distribution of events) was used for the baseline rate, while a spline with 3 degrees of freedom was used for the time-varying effect. In the population comparison, the analyses were fitted in 2 models. In the first model, we performed the analyses without any adjustment. In the second model, we adjusted for sex, age at the index date, country of residence, calendar year of the index date, marital status, highest attained educational level, household income, history of cancer, history of psychiatric disorder, and family history of psychiatric disorder. In the sibling comparison, we also performed the analyses using 2 models. In the unadjusted model, the analyses were stratified by family identifier (mother's and father's identification numbers). In the adjusted model, we additionally adjusted for the same variables as in population comparison except family history of psychiatric disorder.

We then moved on to analyze the effect of parental and child cancer characteristics on the association of interest. In these analyses, we used Cox regression to estimate the average hazard ratio (HR) and 95% confidence interval (CI) of suicide attempt and death by suicide in relation to a child cancer diagnosis, with time since the index date as the underlying time scale and the same adjustment as described in flexible parametric survival models. To assess the effect of parental characteristics, we performed stratified analyses by sex, age at the index date (<40, 40 to 60, or >60 classified to represent younger, middle-age, and older parents), calendar year of the index date (<1990, 1990 to 1999, 2000 to 2009, or ≥2010), country of residence, household

income, marital status, highest attained educational level, history of cancer, history of psychiatric disorder, and family history of psychiatric disorder. As some of the covariates might change secondary to the exposure (i.e., cancer diagnosis of a child), all covariates were ascertained at the index date and used as time-fixed variables in the analyses. We assessed the differences between the stratum-specific HRs by including interaction terms in Cox regression and used Wald test to test the statistical significance of the interaction terms. To assess the effect of child cancer characteristics, we performed subgroup analyses by child age at cancer diagnosis, cancer type (CNS cancer, hematological malignancy, or others), cancer aggressiveness (low, medium, or high), as well as survival status of the child with cancer.

## Sensitivity analyses

We performed several sensitivity analyses to assess the robustness of the findings. To evaluate the effect of lifestyle factors on the association of interest, we performed additional analyses after adjustment for smoking and BMI during pregnancy (only among mothers). To investigate the robustness of multiple imputation, we performed additional analyses using indicator-missing method and complete case method. Finally, as we focused on first-onset suicide attempt, we restricted the main analysis to parents without previous suicide attempt before the index date. In a separate analysis, we constructed an additional cohort of parents (both exposed and unexposed) with at least 1 suicide attempt before the index date to investigate the association of child cancer diagnosis with risk of recurrent suicide attempt, using flexible parametric models.

This study is reported according to the Strengthening the Reporting of Observational Studies in Epidemiology (STROBE) guideline (**S1 Checklist**). All analyses were performed using SAS 9.4 (SAS Institute) and Stata 15.1 (StataCorp LP). Statistical significance was indicated using $P < 0.05$ and Wald test was used to determine $P$ values.

## Ethics statement

The study was approved by the Danish Data Protection Agency (J.nr. 2008-41-2680 and J.nr. 2013-41-2569) and the Ethics Review Board in Stockholm (2016/288-31/1 and 2021–03315).

## Results

### Baseline characteristics

The population comparison consisted of 106,005 exposed parents and 1,060,050 matched unexposed parents, with a median age of 56 at cohort entry and 46.9% male. Exposed parents were more likely to be married and had a lower educational level but a higher prevalence of cancer history, compared with the unexposed parents (**Table 1**). In the sibling comparison, exposed parents were more likely to be female, >60 years, and married, and had a higher prevalence of cancer history, compared with their unexposed siblings.

In the population comparison, the median follow-up time was 7.3 years for the exposed and 7.2 years for the unexposed parents. During follow-up, we observed 613 (IR, 58.8 per 100,000 person-years) and 5,888 (IR, 57.1 per 100,000 person-years) cases of first-onset suicide attempt among the exposed and unexposed parents, respectively. We observed an increased risk of suicide attempt among the exposed parents during the first years after a child cancer diagnosis, but the increased risk did not maintain throughout the follow-up in either the unadjusted (**S2A Fig**) or the adjusted (**Fig 1A**) model. In the sibling comparison, we similarly observed an increased risk of suicide attempt during the first few years after a child cancer diagnosis but not throughout the follow-up, with (**Fig 1B**) or without (**S2B Fig**) multivariable adjustment. In

**Table 1. Baseline characteristics of the study participants.**

| Characteristics | Population comparison | | Sibling comparison | |
|---|---|---|---|---|
| | Exposed parents (N = 106,005) | Matched unexposed parents (N = 1,060,050) | Exposed parents (N = 40,218) | Unexposed full siblings of the exposed parents (N = 64,446) |
| **Male, N (%)** | 49,750 (46.9%) | 497,500 (46.9%) | 19,108 (47.5%) | 32,737 (50.8%) |
| **Median age at cohort entry, years** | 56 | 56 | 51 | 49 |
| **Age at cohort entry, years, N (%)** | | | | |
| <40 | 17,925 (16.9%) | 179,109 (16.9%) | 8,607 (21.4%) | 14,177 (22.0%) |
| 40–60 | 53,057 (50.1%) | 530,454 (50.0%) | 23,507 (58.4%) | 40,091 (62.2%) |
| >60 | 35,023 (33.0%) | 350,487 (33.1%) | 8,104 (20.2%) | 10,178 (15.8%) |
| **Calendar year at cohort entry, N (%)** | | | | |
| 1973–1989 | 7,956 (7.5%) | 79,560 (7.5%) | 2,720 (6.8%) | 4,010 (6.2%) |
| 1990–1999 | 16,159 (15.2%) | 161,590 (15.2%) | 6,479 (16.1%) | 10,426 (16.2%) |
| 2000–2009 | 38,763 (36.6%) | 387,630 (36.6%) | 14,953 (37.2%) | 24,100 (37.4%) |
| 2010–2016 | 43,127 (40.7%) | 431,270 (40.7%) | 16,066 (40.0%) | 25,910 (40.2%) |
| **Country of residence, N (%)** | | | | |
| Sweden | 57,163 (53.9%) | 571,630 (53.9%) | 31,370 (78.0%) | 50,908 (79.0%) |
| Denmark | 48,842 (46.1%) | 488,420 (47.7%) | 8,848 (22.0%) | 13,538 (21.0%) |
| **Educational level, years, N (%)** | | | | |
| 0–9 | 29,629 (28.0%) | 269,323 (25.4%) | 9,506 (23.6%) | 15,556 (24.1%) |
| 10–14 | 56,944 (53.7%) | 568,839 (53.7%) | 23,694 (58.9%) | 38,334 (59.5%) |
| 15- | 17,949 (16.9%) | 205,516 (19.4%) | 6,601 (16.4%) | 9,885 (15.3%) |
| Missing | 1,483 (1.4%) | 16,372 (1.5%) | 417 (1.0%) | 671 (1.0%) |
| **Household income, N (%)** | | | | |
| Below first tertile | 30,772 (29.0%) | 307,335 (29.0%) | 8,221 (20.4%) | 12,901 (20.0%) |
| Between first and second tertile | 27,962 (26.4%) | 270,518 (25.5%) | 12,312 (30.6%) | 19,903 (30.9%) |
| Above second tertile | 45,523 (42.9%) | 464,382 (43.8%) | 18,954 (47.1%) | 30,644 (47.6%) |
| Missing | 1,748 (1.6%) | 17,815 (1.7%) | 731 (1.8%) | 998 (1.6%) |
| **Marital status, N (%)** | | | | |
| Single, widowed, or divorced | 26,534 (25.0%) | 277,507 (26.2%) | 10,159 (25.3%) | 17,154 (26.6%) |
| Married | 59,905 (56.5%) | 586,014 (55.3%) | 17,882 (44.5%) | 27,397 (42.5%) |
| Missing | 19,566 (18.5%) | 196,529 (18.5%) | 12,177 (30.3%) | 19,895 (30.9%) |
| **History of cancer, N (%)** | | | | |
| Yes | 8,268 (7.8%) | 67,868 (6.4%) | 2,170 (5.4%) | 2,770 (4.3%) |
| No | 97,737 (92.2%) | 992,182 (93.6%) | 38,048 (94.6%) | 61,676 (95.7%) |
| **History of psychiatric disorder, N (%)** | | | | |
| Yes | 9,258 (8.7%) | 88,089 (8.3%) | 3,223 (8.0%) | 5,207 (8.1%) |
| No | 96,747 (91.3%) | 971,961 (91.7%) | 36,995 (92.0%) | 59,239 (91.9%) |
| **Family history of psychiatric disorder, N (%)** | | | | |
| Yes | 18,449 (17.4%) | 183,182 (17.3%) | 10,454 (26.0%) | 16,536 (25.7%) |
| No | 87,556 (82.6%) | 876,868 (82.7%) | 29,764 (74.0%) | 47,910 (74.3%) |

the population comparison, we identified 108 (IR, 10.3 per 100,000 person-years) and 1,317 (IR, 12.7 per 100,000 person-years) cases of death by suicide among the exposed and unexposed parents during the follow-up, respectively. Exposed parents had no altered risk of death by suicide, compared to either the unexposed parents (**Figs 1C and S2C**) or their unexposed siblings (**Figs 1D and S2D**), regardless of multivariable adjustment.

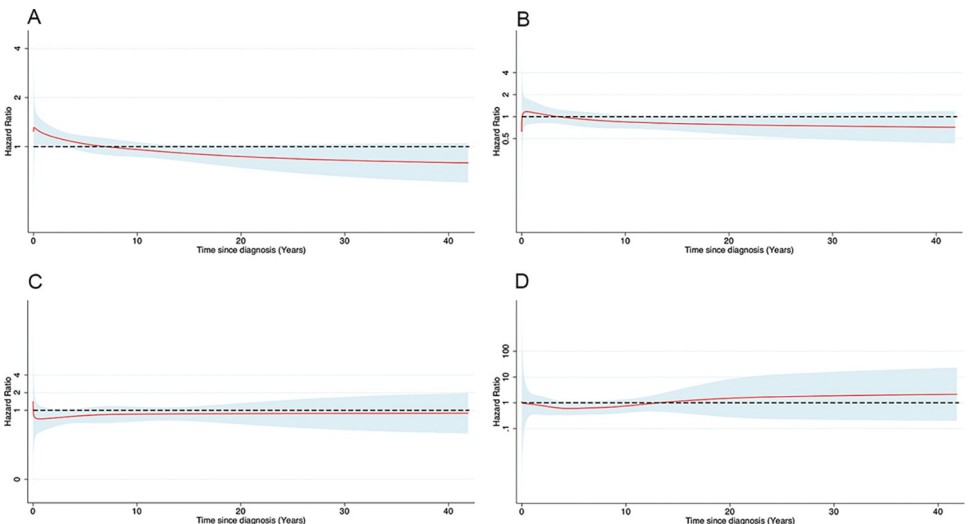

**Fig 1. Adjusted hazard ratio (HR) with 95% confidence interval (CI) of suicide attempt and death by suicide in relation to cancer diagnosis of a child, analyses of population, or sibling comparison.** HR and associated 95% CI were estimated from flexible parametric survival models, allowing the effect of cancer diagnosis of a child to vary over time. A spline with 5 degrees of freedom (4 intermediate knots and 2 knots at each boundary, placed according to quintile distribution of events) was used for the baseline rate, while a spline with 3 degrees of freedom was used for the time-varying effect. (**A**) and (**C**) were adjusted for sex, age at cohort entry, country of residence, calendar year of cohort entry, marital status, the highest attained educational level, household income, history of cancer, history of psychiatric disorder, and family history of psychiatric disorder. (**B**) and (**D**) were adjusted for sex, age at cohort entry, country of residence, calendar year of cohort entry, marital status, the highest attained educational level, household income, history of cancer, and history of psychiatric disorder. The analyses of flexible parametric survival models were performed on 5 imputed datasets, and HRs were obtained from combination of each dataset using Rubin's rule. (**A**) Suicide attempt in population comparison. (**B**) Suicide attempt in sibling comparison. (**C**) Death by suicide in population comparison. (**D**) Death by suicide in sibling comparison.

Given the null association for death by suicide in both the population and sibling comparisons as well as the smaller number of suicide attempts in the sibling comparison compared to the population comparison, we focused on suicide attempt and the population comparison in the following analyses.

As the average HRs of suicide attempt were similar between the unadjusted and adjusted models, we present HRs with multivariable adjustment (Table 2). For instance, the mean HR of suicide attempt was 1.15 (95% CI, [1.03, 1.28]; $p = 0.01$) during the first 7 years after a child cancer diagnosis and 0.86 (95% CI, [0.75, 0.98]; $p = 0.03$) thereafter. Such increased risk was mainly attributable to mothers, younger parents (60 or younger), earlier calendar years (before 2000), as well as parents without a history of cancer, a history of psychiatric disorder, or a family history of psychiatric disorder. The lack of increased risk of suicide attempt thereafter was, on the other hand, mainly attributable to fathers, older parents, as well as parents without a history of cancer or a family history of psychiatric disorder.

In the analysis of child cancer characteristics, we focused on suicide attempt and found that the increased risk of suicide attempt soon after a child's cancer diagnosis was mainly observed in relation to a child cancer diagnosis at 18 or younger (HR, 1.25; 95% CI, [1.08, 1.46]; $p = 0.004$), a child diagnosis of hematological malignancy (HR, 1.26; 95% CI, [1.02, 1.55]; $p = 0.03$), a child diagnosis of highly aggressive cancer (HR, 1.60; 95%CI, [1.05, 2.43]; $p = 0.03$), and when the child died from the cancer (HR, 1.63; 95% CI, [1.29, 2.06]; $p < 0.001$) (Table 3).

Further adjustment for smoking (HR, 1.20, 95% CI, [1.04, 1.39]; $p = 0.01$) or BMI (HR, 1.17, 95% CI, [0.99, 1.37]; $p = 0.06$) during pregnancy as well as using complete case method

**Table 2. IR (per 100,000 person-years) and HR with 95% CI of suicide attempt following a child cancer diagnosis—Analysis of the population comparison.**

| Characteristics | Parents of cancer-free children | | Parents of children with cancer | | | | | | | |
|---|---|---|---|---|---|---|---|---|---|---|
| | Entire follow-up | | Entire follow-up | | First 7 years of follow-up | | | >7 years of follow-up | | |
| | No. of cases | IR | No. of cases | IR | Unadjusted HR (95% CI)[a]; P value | Adjusted HR (95% CI)[b]; P value | P for interaction | Unadjusted HR (95% CI)[a]; P value | Adjusted HR (95% CI)[b]; P value | P for interaction |
| **Overall** | 5,888 | 57.1 | 613 | 58.8 | **1.17 (1.11, 1.23); P < 0.001** | **1.15 (1.03, 1.28); P = 0.01** | | **0.87 (0.82, 0.92); P < 0.001** | **0.86 (0.75, 0.98); P = 0.03** | |
| **Sex** | | | | | | | 0.15 | | | 0.17 |
| Male | 2,531 | 52.4 | 235 | 48.1 | 1.04 (0.97, 1.13); P = 0.27 | 1.04 (0.88, 1.24); P = 0.63 | | **0.76 (0.69, 0.84); P < 0.001** | **0.77 (0.62, 0.95); P = 0.02** | |
| Female | 3,357 | 61.2 | 378 | 68.2 | **1.27 (1.19, 1.35); P < 0.001** | **1.23 (1.07, 1.41); P < 0.001** | | 0.95 (0.88, 1.02); P = 0.15 | 0.93 (0.79, 1.10); P = 0.40 | |
| **Age at cohort entry, years** | | | | | | | 0.04 | | | 0.11 |
| <40 | 2,765 | 82.1 | 297 | 87.1 | **1.25 (1.15, 1.35); P < 0.001** | **1.27 (1.06, 1.52); P = 0.009** | | 0.95 (0.88, 1.02); P = 0.15 | 0.96 (0.81, 1.12); P = 0.58 | |
| 40–60 | 2,583 | 47.9 | 273 | 50.2 | **1.22 (1.14, 1.30); P < 0.001** | **1.18 (1.02, 1.36); P = 0.03** | | **0.75 (0.67, 0.83); P < 0.001** | **0.73 (0.57, 0.93); P = 0.01** | |
| >60 | 540 | 34.6 | 43 | 27.3 | **0.80 (0.70, 0.94); P = 0.005** | 0.79 (0.56, 1.09); P = 0.15 | | **0.54 (0.35, 0.85); P = 0.008** | 0.53 (0.19, 1.45); P = 0.15 | |
| **Calendar year at cohort entry** | | | | | | | 0.18 | | | 0.73 |
| 1973–1989 | 1,579 | 68.9 | 166 | 71.5 | **1.34 (1.19, 1.50); P < 0.001** | **1.32 (1.02, 1.71); P < 0.001** | | **0.91 (0.83, 0.99); P = 0.04** | 0.90 (0.73, 1.11); P = 0.32 | |
| 1990–1999 | 1,860 | 61.4 | 190 | 62.2 | **1.32 (1.20, 1.44); P < 0.001** | **1.31 (1.06, 1.61); P < 0.001** | | **0.80 (0.73, 0.89); P < 0.001** | 0.80 (0.65, 1.00); P = 0.05 | |
| 2000–2009 | 1,893 | 51.1 | 199 | 53.1 | **1.08 (1.00, 1.17); P = 0.04** | 1.06 (0.89, 1.25); P = 0.14 | | 0.92 (0.80, 1.05); P = 0.22 | 0.91 (0.67, 1.23); P = 0.54 | |
| 2010–2016 | 556 | 43.1 | 58 | 44.6 | 1.04 (0.92, 1.18); P = 0.51 | 1.01 (0.77, 1.32); P = 0.93 | | - | - | |
| **Country of residence** | | | | | | | 0.43 | | | 0.36 |
| Sweden | 3,101 | 54.6 | 320 | 55.9 | **1.21 (1.14, 1.30); P < 0.001** | **1.19 (1.03, 1.38); P < 0.001** | | **0.82 (0.75, 0.89); P < 0.001** | **0.82 (0.68, 0.98); P = 0.03** | |
| Denmark | 2,787 | 60.1 | 293 | 62.4 | **1.12 (1.05, 1.20); P = 0.001** | 1.10 (0.94, 1.28); P = 0.25 | | 0.93 (0.85, 1.01); P = 0.10 | 0.91 (0.75, 1.11); P = 0.35 | |
| **Educational level, years** | | | | | | | 0.79 | | | 0.66 |
| 0–9 | 2,252 | 81.3 | 249 | 82.6 | **1.11 (1.03, 1.20); P = 0.008** | 1.12 (0.94, 1.33); P = 0.21 | | **1.11 (1.03, 1.20); P = 0.008** | 0.92 (0.75, 1.12); P = 0.40 | |
| 10–14 | 2,904 | 52.3 | 294 | 52.4 | **1.17 (1.09, 1.25); P < 0.001** | 1.16 (0.99, 1.34); P = 0.06 | | **0.80 (0.73, 0.87); P < 0.001** | **0.80 (0.66, 0.98); P = 0.03** | |
| ≥15 | 582 | 31.5 | 58 | 35.0 | **1.22 (1.05, 1.41); P = 0.008** | 1.26 (0.91, 1.76); P = 0.17 | | 0.94 (0.77, 1.16); P = 0.57 | 0.97 (0.61, 1.55); P = 0.89 | |
| **Household income** | | | | | | | 0.76 | | | 0.88 |
| Below the lowest tertile | 1,897 | 89.7 | 200 | 96.2 | **1.13 (1.05, 1.22); P = 0.001** | 1.10 (0.93, 1.30); P = 0.28 | | 0.94 (0.83, 1.06); P = 0.29 | 0.90 (0.68, 1.18); P = 0.44 | |
| Between the lowest and highest tertile | 1,745 | 65.6 | 192 | 68.6 | **1.19 (1.09, 1.30); P < 0.001** | 1.17 (0.96, 1.43); P = 0.12 | | **0.87 (0.80, 0.96); P = 0.005** | 0.87 (0.70, 1.08); P = 0.21 | |
| Above the highest tertile | 1,880 | 37.8 | 186 | 37.5 | **1.20 (1.10, 1.31); P < 0.001** | 1.19 (0.98, 1.44); P = 0.08 | | **0.82 (0.74, 0.91); P < 0.001** | 0.83 (0.66, 1.04); P = 0.11 | |
| **Marital status** | | | | | | | 0.34 | | | 0.88 |
| Single, widowed, or divorced | 2,685 | 96.8 | 261 | 100.3 | **1.13 (1.05, 1.21); P < 0.001** | 1.09 (0.93, 1.27); P = 0.25 | | **0.87 (0.80, 0.95); P = 0.003** | 0.85 (0.69, 1.05); P = 0.13 | |

*(Continued)*

**Table 2.** (Continued)

| Characteristics | Parents of cancer-free children | | Parents of children with cancer | | | | | | | |
|---|---|---|---|---|---|---|---|---|---|---|
| | Entire follow-up | | Entire follow-up | | First 7 years of follow-up | | | >7 years of follow-up | | |
| | No. of cases | IR | No. of cases | IR | Unadjusted HR (95% CI)[a]; P value | Adjusted HR (95% CI)[b]; P value | P for interaction | Unadjusted HR (95% CI)[a]; P value | Adjusted HR (95% CI)[b]; P value | P for interaction |
| Married | 2,722 | 42.9 | 303 | 46.4 | **1.25 (1.16, 1.33); P < 0.001** | **1.21 (1.04, 1.40); P = 0.02** | | **0.89 (0.82, 0.97); P = 0.005** | 0.87 (0.72, 1.04); P = 0.14 | |
| **History of cancer** | | | | | | | 0.28 | | | 0.42 |
| Yes | 175 | 50.8 | 22 | 50.0 | 0.90 (0.71, 1.13); P = 0.37 | 0.89 (0.53, 1.50); P = 0.67 | | 1.30 (0.88, 1.92); P = 0.19 | 1.17 (0.49, 1.80); P = 0.72 | |
| No | 5,731 | 57.7 | 591 | 59.2 | **1.18 (1.13, 1.24); P < 0.001** | **1.16 (1.04, 1.30); P = 0.007** | | **0.86 (0.81, 0.92); P < 0.001** | **0.86 (0.75, 0.98); P = 0.02** | |
| **History of psychiatric disorder** | | | | | | | 0.27 | | | 0.87 |
| Yes | 1,829 | 317.0 | 193 | 317.5 | 1.07 (0.99, 1.16); P = 0.09 | 1.07 (0.90, 1.27); P = 0.44 | | **0.84 (0.74, 0.96); P = 0.01** | 0.85 (0.63, 1.14); P = 0.27 | |
| No | 4,059 | 41.7 | 420 | 42.8 | **1.21 (1.14, 1.28); P < 0.001** | **1.20 (1.05, 1.37); P = 0.009** | | **0.87 (0.81, 0.93); P < 0.001** | 0.86 (0.74, 1.00); P = 0.06 | |
| **Family history of psychiatric disorder** | | | | | | | 0.15 | | | 0.38 |
| Yes | 1,143 | 82.9 | 113 | 55.3 | 1.00 (0.90, 1.12); P = 1.00 | 0.98 (0.76, 1.25); P = 0.86 | | 0.98 (0.86, 1.13); P = 0.82 | 0.98 (0.72, 1.34); P = 0.90 | |
| No | 4,745 | 53.1 | 500 | 82.0 | **1.22 (1.15, 1.28); P < 0.001** | **1.19 (1.06, 1.34); P = 0.003** | | **0.85 (0.79, 0.91); P < 0.001** | **0.84 (0.72, 0.97); P = 0.02** | |

[a]Without any adjustment. Time since cohort entry was used as the underlying time scale. The analyses were performed on 5 imputed datasets, and HRs were obtained from combination of each dataset using Rubin's rule.

[b]Adjusted for sex, age at cohort entry, country of residence, calendar year at cohort entry, marital status, the highest attained educational level, household income, history of cancer, history of psychiatric disorder, and family history of psychiatric disorder. Time since cohort entry was used as the underlying time scale. The analyses were performed on 5 imputed datasets, and HRs were obtained from combination of each dataset using Rubin's rule.

Bold confidence intervals and p-values refer to statistically significant results.

CI, confidence interval; HR, hazard ratio; IR, incidence rate.

(HR, 1.16, 95% CI, [1.04, 1.29]; $p = 0.01$) or indicator-missing method (HR, 1.15, 95% CI, [1.03, 1.28]; $p = 0.01$) for missing imputation did not change the results substantially. In the sibling comparison, the mean HR of suicide attempt was 1.45 (95% CI, [1.00, 2.10]; $p = 0.049$) during the first 3.5 years after a child cancer diagnosis and 0.89 (95% CI, [0.70, 1.13]; $p = 0.34$) thereafter. Finally, in separate analyses of parents with at least 1 suicide attempt before the index date, we observed an increased risk of recurrent suicide attempt during the first 2 years after a child cancer diagnosis but not thereafter (**S3A and S3B Fig**), and there was no increased risk of death by suicide any time after a child cancer diagnosis (**S3C and S3D Fig**).

## Discussion

In this binational cohort study, we observed an increased risk of parental suicide attempt during the first years after a child cancer diagnosis, particularly when the child was diagnosed at 18 or younger or when the child was diagnosed with an aggressive or fatal cancer. There was no increased risk of parental suicide attempt later during follow-up, and there was no altered risk of parental death by suicide any time after a child cancer diagnosis.

**Table 3. HR with 95% CI of suicide attempt during the first 7 years after a child cancer diagnosis—Subgroup analysis of population comparison by cancer characteristics and calendar period.**

| Cancer characteristics | HR (95% CI)[a] | P value |
|---|---|---|
| **Age at cancer diagnosis** | | |
| ≤18 years | **1.25 (1.08, 1.46)** | **0.004** |
| >18 years | 1.07 (0.92, 1.24) | 0.36 |
| **Cancer type** | | |
| CNS tumor | 1.18 (0.91, 1.54) | 0.21 |
| Hematological malignancy | **1.26 (1.02, 1.55)** | **0.03** |
| Other cancer types | 1.10 (0.96, 1.26) | 0.16 |
| **Cancer aggressiveness[b]** | | |
| Low | 1.10 (0.89, 1.37) | 0.37 |
| Medium | 1.13 (1.00, 1.28) | 0.05 |
| High | **1.60 (1.05, 2.43)** | **0.03** |
| **Survival status** | | |
| Surviving | 1.07 (0.95, 1.21) | 0.25 |
| Deceased | **1.63 (1.29, 2.06)** | **<0.001** |

[a]Cox models were adjusted for sex, age at cohort entry, country of residence, calendar year at cohort entry, marital status, the highest attained educational level, household income, history of cancer, history of psychiatric disorder, and family history of psychiatric disorder. Time since cohort entry was used as the underlying time scale. The analyses of Cox models were performed on 5 imputed datasets, and HRs were obtained from combination of each dataset using Rubin's rule.

[b]Cancer aggressiveness was defined according to 5-year survival rate of each cancer type (https://www-dep.iarc.fr/nordcan/english/frame.asp.). Low aggressiveness of cancer type includes breast cancer, prostate cancer, non-melanoma skin cancer, melanoma, corpus uteri cancer, and thyroid cancer. High aggressiveness of cancer type includes lung cancer, esophagus cancer, liver cancer, pancreatic cancer, ovary cancer, and unknown/ill-defined cancer. Medium aggressiveness of cancer type includes other cancer types not included in low aggressiveness and high aggressiveness.

Bold confidence intervals and *p*-values refer to statistically significant results.

CI, confidence interval; CNS, central nervous system; HR, hazard ratio.

Relatively little is known regarding the risk of suicidal behavior among parents of children with cancer. The few studies assessing the risk of death by suicide among parents of a child with cancer have so far yielded inconsistent results [5,9,10]. This may in part be owing to the relatively limited sample size and the short follow-up of the previous studies, considering the low absolute risk of death by suicide. In line with 2 of these studies [5,10], we found in the present study little evidence for an association between cancer diagnosis of a child and risk of death by suicide among parents. Our study extends, on the other hand, the existing knowledge base by demonstrating an increased risk of parental suicide attempt during the first years after a child cancer diagnosis, but not thereafter.

Having a child with cancer is a severely stressful life event [1]. The observed increased risk of parental suicide attempt within the first years after a child cancer diagnosis might be attributed to the psychological distress of having a child receiving a cancer diagnosis and the subsequent treatment of cancer. In addition, changes in financial [23] and marital status [24], as potential subsequence of a child cancer diagnosis, might also contribute to the observed risk increase. The increased risk was mainly attributable to mothers and younger parents. The former is likely because females might be more susceptible to psychological distress than males [25,26]. The latter, on the other hand, corroborates the observation that the increased risk of suicide attempt during the first years after a child cancer diagnosis was mainly observed for a

child cancer diagnosed at age 18 or younger. In addition, the risk increase of suicide attempt was also stronger for a child diagnosis of cancer with poor prognosis and when the child died from cancer.

In contrast to an increased risk of parental suicide attempt during the first years after a child cancer diagnosis, we observed no increased risk of parental suicide attempt later during the follow-up and no altered risk of parental death by suicide any time after a child cancer diagnosis. The exact mechanisms underlying such contrasting findings are yet known. However, one might speculate that parenting a child with cancer might eventually increase the resilience and ability of the parents to adapt to challenges and better deal with future stress [27,28]. Another explanation might be the differential clinical characteristics and risk factors between suicide attempt and death by suicide [29,30]. For instance, some might conduct self-harm to cope with extreme negative emotions, with however no clear suicidal intent. Regardless, the different risk patterns between suicide attempt and death by suicide needs to be validated in future studies with independent study populations.

The major strengths of the study are the large sample size and the long follow-up. Thanks to the population-based design, the complete follow-up due to register linkage, and the independent and prospective ascertainment of child cancer diagnosis as well as parental suicidal behavior, concerns of information and selection biases should have been minimized. The positive predictive value (PPV) of using register-based definitions of suicide attempt was shown to be high in the Swedish and Danish registers (above 80%) [18,31]. The accuracy of register-based definitions of death by suicide is also high, with a reported 81% and 90% agreement after expert reevaluation in Sweden and Denmark, respectively [32]. Further, the multivariable adjustment in the main and sensitivity analyses as well as the sibling comparison enabled thorough control for confounding. Our study also has limitations. First, although we tried our best to control for the effect of some lifestyle factors (for instance, smoking and BMI during pregnancy), the impact for other lifestyle factors was not addressed. However, to qualify as confounders, such factors have been common causes for both child cancer diagnosis and parental suicidal behavior. The diminished estimates observed in the sibling comparison could partly speak for residual confounding due to shared familial factors. However, as siblings of the exposed individuals also experienced, to different extent, the psychological distress of having a niece or nephew with a cancer diagnosis, the results of the sibling comparison may also represent an underestimation of the real association (i.e., overmatching). Second, given a lack of information on prognostic markers of cancer (for instance, cancer stage at diagnosis), we classified cancer aggressiveness based on cancer type, which might have led to some misclassification. Third, there is a concern of surveillance bias, assuming parents of children with cancer might have a greater access to healthcare, leading to a greater possibility of being recorded for suicide attempt in the healthcare system, or, alternatively, a lower chance of presenting themselves to healthcare in case of ill health, due to psychological distress or the demanding parenting of a child with cancer. Fourth, although our study had a binational study population and a large sample size, we had, regardless, relatively limited statistical power in the analysis of death by suicide given the rareness of this event. Fifth, given that parents of children with cancer might have a higher mortality than others, competing risk due to other causes of death might have diluted the observed positive association. Sixth, as the presence of other children has been shown to alleviate grief of parents undergoing stressful life events, for instance, bereavement of a child due to death [33,34], the association of child cancer diagnosis with parental suicidal behavior might vary for parents with or without other children. Future studies are therefore warranted to test such a hypothesis. Finally, the generalizability of our findings to other countries with different healthcare system, sociocultural context, and prevalence of cancer and suicidal behavior from those of Denmark and Sweden needs caution.

In conclusion, there was an increased risk of parental suicide attempt during the first years after the diagnosis of a child cancer, particularly when the child was diagnosed at age 18 or below, or when the child was diagnosed with a cancer of poor prognosis, but not thereafter. There was no altered risk of parental death by suicide any time after a child cancer diagnosis. Our findings suggest the need of extended clinical awareness for the risk of suicide attempt among parents of children with cancer, primarily during the first years after cancer diagnosis. Vigilance and interventions for severe psychological distress and psychiatric symptoms might be important and effective.

## Supporting information

**S1 Fig. Flow chart of the study design.**
(PDF)

**S2 Fig. Unadjusted hazard ratio (HR) with 95% confidence interval (CI) of suicide attempt and death by suicide in relation to cancer diagnosis of a child, analyses of population, or sibling comparison.** HR and associated 95% CI were estimated from flexible parametric survival models, allowing the effect of cancer diagnosis of a child to vary over time. A spline with 5 degrees of freedom (4 intermediate knots and 2 knots at each boundary, placed according to quintile distribution of events) was used for the baseline rate, while a spline with 3 degrees of freedom was used for the time-varying effect. (**A**), (**B**), (**C**), and (**D**) were without any adjustment. The analyses of flexible parametric survival models were performed on 5 imputed datasets, and HRs were obtained from combination of each dataset using Rubin's rule. (**A**) Suicide attempt in population comparison. (**B**) Suicide attempt in sibling comparison. (**C**) Death by suicide in population comparison. (**D**) Death by suicide in sibling comparison.
(PDF)

**S3 Fig. Hazard ratio (HR) with 95% confidence interval (CI) of suicide attempt and suicide by death in relation to cancer diagnosis of a child, analyses among parents with a history of suicide attempt.** HR and associated 95% CI were estimated from flexible parametric survival models, allowing the effect of cancer diagnosis of a child to vary over time. A spline with 5 degrees of freedom (4 intermediate knots and 2 knots at each boundary, placed according to quintile distribution of events) was used for the baseline rate, while a spline with 3 degrees of freedom was used for the time-varying effect. (**A**) and (**C**) were without any adjustment. (**B**) and (**D**) were adjusted for sex, age at cohort entry, country of residence, calendar year at cohort entry, marital status, the highest attained education, personal income, history of cancer, history of psychiatric disorder, and family history of psychiatric disorder. The analyses of flexible parametric survival models were performed on 5 imputed datasets, and HRs were obtained from combination of each dataset using Rubin's rule. (**A**) Suicide attempt in unadjusted model. (**B**) Suicide attempt in adjusted model. (**C**) Death by suicide in unadjusted model. (**D**) Death by suicide in adjusted model.
(PDF)

**S1 Table. Danish and Swedish revisions of the International Classification of Diseases (ICD) codes for cancer subtypes in the Danish and Swedish Cancer Registers.**
(PDF)

**S2 Table. Danish and Swedish revisions of the International Classification of Diseases (ICD) codes for suicide attempt and psychiatric disorder in the Patient Register and the Psychiatric Central Register in Denmark and Patient Register in Sweden, and death by**

suicide in the Danish Causes of Death Register and the Swedish Causes of Death Register.
(PDF)

**S1 Checklist. STROBE Statement—Checklist of items that should be included in reports of observational studies.**
(PDF)

**S1 Text. Analysis plan—Suicidal behavior among parents of children with cancer.**
(PDF)

**S2 Text. Supplementary methods of multiple imputation.**
(PDF)

## Author Contributions

**Conceptualization:** Qianwei Liu, Krisztina D. László, Jiong Li, Fang Fang.

**Data curation:** Jiong Li.

**Formal analysis:** Qianwei Liu.

**Funding acquisition:** Qianwei Liu, Krisztina D. László, Dang Wei, Jiong Li, Fang Fang.

**Investigation:** Qianwei Liu.

**Methodology:** Qianwei Liu, Dang Wei, Fang Fang.

**Resources:** Krisztina D. László, Jiong Li.

**Supervision:** Jiong Li, Fang Fang.

**Visualization:** Qianwei Liu.

**Writing – original draft:** Qianwei Liu.

**Writing – review & editing:** Qianwei Liu, Krisztina D. László, Dang Wei, Fen Yang, Katja Fall, Unnur Valdimarsdóttir, Maria Feychting, Jiong Li, Fang Fang.

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
