## [Editor Report · Decision Letter 0]

29 Mar 2023

Dear Dr Liu, 

Thank you for submitting your manuscript entitled "Suicidal attempt and completed suicide among parents of children with cancer: a population-based study in Denmark and Sweden" for consideration by PLOS Medicine.

Your manuscript has now been evaluated by the PLOS Medicine editorial staff and I am writing to let you know that we would like to send your submission out for external peer review.

Please re-submit your manuscript within two working days, i.e. by Mar 31 2023 11:59PM.

Kind regards,

Philippa Dodd, MBBS MRCP PhD

PLOS Medicine

---

## [Decision Letter · Decision Letter 1]

3 Oct 2023

Dear Dr. Liu,

Thank you very much for submitting your manuscript "Suicidal attempt and completed suicide among parents of children with cancer: a population-based study in Denmark and Sweden" (PMEDICINE-D-23-00808R1) for consideration at PLOS Medicine. 

[LINK]

In light of these reviews, I am afraid that we will not be able to accept the manuscript for publication in the journal in its current form, but we would like to consider a revised version that addresses the reviewers' and editors' comments. Obviously we cannot make any decision about publication until we have seen the revised manuscript and your response, and we plan to seek re-review by one or more of the reviewers. 

We expect to receive your revised manuscript by Oct 24 2023 11:59PM. Please email us (plosmedicine@plos.org) if you have any questions or concerns.

We look forward to receiving your revised manuscript. 

Sincerely,

Philippa Dodd, MBBS MRCP PhD

PLOS Medicine

plosmedicine.org

COMMENTS FROM THE EDITORS

GENERAL

Please respond to all editor and reviewer comments detailed below in full.

Please include line numbers starting at line 1 of the abstract and in continuous sequence thereafter and throughout.

Please ensure that the study is reported according to the STROBE guideline, and include the completed STROBE checklist as Supporting Information. Please add the following statement, or similar, to the Methods: "This study is reported as per the Strengthening the Reporting of Observational Studies in Epidemiology (STROBE) guideline (S1 Checklist)."

When completing the checklist, please refer to section and paragraph numbers, rather than page or line numbers as these often change in the event of publication.

* The editorial team agree that your manuscript addresses an important subject matter. However, the implications of your findings going forwards should be better discussed. * 

** The editorial team agree with reviewer #3’s concerns (please see below) regarding stratification according to sub-categories of childhood malignancy and require detailed justification for this. **

*** We suggest tempering the claim of ‘decreased’ risks of suicide after 7 years and simply stating that increased risk is not maintained beyond 7 years ***

DATA AVAILABILITY STATEMENT

Thank you for including a detailed statement. Please provide a URL or email address for data enquiries to each data provider - Statistics Denmark, Swedish National Board of Health and Welfare, and Statistics Sweden. Please note that the contact cannot be a study author.

COMPETING INTERESTS

All authors must declare their relevant competing interests per the PLOS policy, which can be seen here:

https://journals.plos.org/plosmedicine/s/competing-interests

For authors with ties to industry, please indicate whether any of the interests has a financial stake in the results of the current study.

HIGHLIGHTS – page 2 – please remove this section.

ABSTRACT

Please structure your abstract using the PLOS Medicine headings (Background, Methods and Findings, Conclusions).

Please combine the Methods and Findings sections into one section, “Methods and findings”.

Abstract Background: Provide wider context of why the study is important. The final sentence should clearly state the study question.

Abstract Methods and Findings:

Please ensure that all numbers presented in the abstract are present and identical to numbers presented in the main manuscript text.

Please provide a (brief) summary of aggregate demographic details of the study population – number of participants, sex, age (and range) and so on.

Please include the length of follow up (in mean, SD, and range) and main outcome measures. 

Please quantify the main results with 95% CIs and p values. When reporting p values please report as p<0.001 and where higher the exact p value as p=0.002, for example. When reporting CIs please separate upper and lower bounds with commas as opposed to hyphens as the latter can be confused with reporting of negative values.

Suggest reporting statistical information as follows, ‘(HR, 1.15; 95%CI [1.03,1.28]; p </=’

Please include any important dependent variables that are adjusted for in the analyses.

Please include the absolute risk(s) of relevant outcomes (including NNT or NNH where appropriate), not just relative risks or correlation coefficients. (example for absolute risks: PMID: 28399126). 

In the last sentence of the Abstract Methods and Findings section, please describe the main limitation(s) of the study's methodology.

Abstract Conclusions:

Please address the study implications without overreaching what can be concluded from the data; the phrase "In this study, we observed ..." may be useful.

Please interpret the study based on the results presented in the abstract, emphasizing what is new/what this adds/how this could influence change, without overstating your conclusions.

Please avoid vague statements such as "these results have major implications for policy/clinical care". Mention only specific implications substantiated by the results.

Please ensure to avoid any assertions of primacy ("We report for the first time....")

AUTHOR SUMMARY

At this stage, we ask that you include a short, non-technical Author Summary of your research to make findings accessible to a wide audience that includes both scientists and non-scientists. The authors summary should consist of 2-3 succinct bullet points under each of the following headings:

• Why Was This Study Done? Authors should reflect on what was known about the topic before the research was published and why the research was needed.

• What Did the Researchers Do and Find? Authors should briefly describe the study design that was used and the study’s major findings. Do include the headline numbers from the study, such as the sample size and key findings. 

• What Do These Findings Mean? Authors should reflect on the new knowledge generated by the research and the implications for practice, research, policy, or public health. Authors should also consider how the interpretation of the study’s findings may be affected by the study limitations. In the final bullet point of ‘What Do These Findings Mean?’, please describe the main limitations of the study in non-technical language.

The Author Summary should immediately follow the Abstract in your revised manuscript. This text is subject to editorial change and should be distinct from the scientific abstract. Please see our author guidelines for more information: https://journals.plos.org/plosmedicine/s/revising-your-manuscript#loc-author-summary

INTRODUCTION

The current introduction is rather brief. Please address past research and explain the need for and potential importance of your study. Indicate whether your study is novel and how you determined that. If there has been a systematic review of the evidence related to your study (or you have conducted one), please refer to and reference that review and indicate whether it supports the need for your study.

Please conclude the Introduction with a clear description of the study question or hypothesis, as in your current version.

METHODS and RESULTS

Please see comments below from reviewer #1 (statistical reviewer) regarding the approach to your analyses.

Please see reviewer #3 comments regarding sub-categorization of child hood malignancy which we require detailed justification for.

Did your study have a prospective protocol or analysis plan? Please state this (either way) early in the Methods section.

For all observational studies, we request that in the manuscript text, authors please indicate: 

(1) the specific hypotheses you intended to test, 

(2) the analytical methods by which you planned to test them, 

(3) the analyses you actually performed, and 

(4) when reported analyses differ from those that were planned, transparent explanations for differences that affect the reliability of the study's results. If a reported analysis was performed based on an interesting but unanticipated pattern in the data, please be clear that the analysis was data-driven.

Please report the number of patients and dates of recruitment, and account for all methods used in your study.

As for the abstract, please quantify the main results with 95% CIs and p values. When reporting p values please report as p<0.001 and where higher the exact p value as p=0.002, for example. When reporting CIs please separate upper and lower bounds with commas as opposed to hyphens as the latter can be confused with reporting of negative values.

Suggest reporting statistical information as follows, ‘(HR, 1.15; 95%CI [1.03,1.28]; p </=’

When a p value is given, please specify the statistical test used to determine it.

Page 6 – ‘cancer child’ please refrain from using this term, suggest ‘child with a cancer diagnosis’ or similar. Please check and amend throughout all sections of the manuscript including the supporting files where relevant.

Page 11 – please define ‘IR’ at first use for the reader – apologies if I have missed it.

Please define the length of follow up (eg, in mean, SD, and range).

At the beginning of your results section (PDF page 11) please include a brief summary of the aggregate demographic details of the study population – number of participants, sex, age (and range) and so on.

TABLES

Table 2

I found this table a little confusing. It wasn’t entirely clear (without referring to the text) how participant numbers apply to the results presented. Please ensure that the table can be interpreted by the reader without the need to refer to the text. Please indicate what the different results in bold refer to.

Please also see the reviewer #3 comments (below) regarding table 2 in relation to the figures. 

Please separate upper and lower bounds of CIs with commas as opposed to hyphens as the later can be confused with reporting of negative values. 

Thank you for indicating that your results are adjusted. To help facilitate transparent data reporting please also include unadjusted analyses for comparison.

Table 3

Where reporting 95% CIs please also report p values. If not reporting p values for the purpose of transparent data reporting, please clearly state the reasons why not.

FIGURES

Figure 1 A-D – thank you for indicating your analyses are adjusted and detailing the factors you have adjusted for. To help facilitate transparent data reporting please also include the unadjusted analyses for comparison. In the caption please clearly define ‘df’ for the reader (degrees of freedom?). 

Please clearly define the meaning of the different colour lines on the graphs. 

Suggest perhaps swapping these colours as the reader is drawn to the colour first but the coloured line this is not the primary feature (the black line should be). Suggest also making the line at 1 dashed to further minimize its prominence. 

DISCUSSION

Please present and organize the Discussion as follows: a short, clear summary of the article's findings; what the study adds to existing research and where and why the results may differ from previous research; strengths and limitations of the study; implications and next steps for research, clinical practice, and/or public policy; one-paragraph conclusion. Please avoid the use of sub-headings such that the discussion reads as continuous prose.

Please remove the funding and disclosure statements from the end of the main manuscript and include only in the masncuript submission form. In the event of publication they will be compiled as metadata.

REFERENCES

For in-text reference callouts please place citations is square brackets and preceding punctuation. For example, [1,4,6].

In the bibliography please list up to but no more than 6 author names followed by et al in the event that more than 6 authors contribute to an individual study. Please ensure that journal name abbreviations are those listed in National Center for Biotechnology Information (NCBI) databases. 

Please see our website for other reference guidelines https://journals.plos.org/plosmedicine/s/submission-guidelines#loc-references

SUPPORTING INFORMATION

Please cite your Supporting Information as outlined here: https://journals.plos.org/plosmedicine/s/supporting-information

In the published article, supporting information files are accessed only through a hyperlink attached to the captions. For this reason, you must list captions at the end of your manuscript file. You may include a caption within the supporting information file itself, as long as that caption is also provided in the manuscript file. Do not submit a separate caption file.

As above, please include the completed STROBE checklist as Supporting Information. Please ensure to refer to section and paragraph numbers as opposed to page or line numbers as these often change in the event of publication.

SUPPLEMENTARY FIGURES

As for the main manuscript suggest revising the graphs.

COMMENTS FROM THE ACADEMIC EDITOR

I looked through the paper in detail and agree that it seems important and well-conducted. My main observation was that the difference between the findings for suicide attempt and death by suicide (preferred terminology to ‘completed suicide’) might be explained by ascertainment bias. Many suicide attempts will not come to the attention of the health service and be picked up in the registry. I thought it was plausible that parents of a child with cancer would be having more contact with health services (albeit not for themselves) and that this could facilitate their presentation to services with a suicide attempt (e.g. family support may mean the parent has the opportunity to disclose). That said there is consistency of findings across the different analyses that support their veracity. Some minor other points:

- The term ‘child’ is accurate but is somehow misleading. It is really being used in the term of offspring/son/daughter and not someone who is a ‘child’ i.e. below the age of 18 years. This becomes apparent as you read the paper but is not what I expected from the title.

- In the methods the term ‘a cancer child’ needs to be amended e.g. to ‘a child with cancer’.

- In the discussions, the increased risk of suicide attempts in mothers is explained as “likely because mothers tend to have a stronger emotional link with their children”. That seems highly speculative and potentially offensive so would need to be well-supported by citations evidence (that I don’t think exists).

Comments from the reviewers:

Reviewer #1: This is a well-conducted population-based study in Denmark and Sweden on suicidal attempt and completed suicide among parents of children with cancer. The study design, datasets, statistical methods and analyses, and presentation (tables and figures) and interpretation of the results are mostly adequate. However, there are still a few statistical issues needing attention.

1) On page 11, it's said "We observed an increased risk of suicidal attempt among the exposed parents during the first seven years after a child cancer diagnosis but a decreased risk thereafter (Figure 1A)". However, Figure 1A shows only dereasing HRs - it's one way traffic. Where does this increase and then decrease come from? The 7 years cut off is a bit arbitrary and perhaps more solid and robust evidence are needed to support this. On the other hand, as we can see the proportional hazard assumption doesn't hold in this study. A time-dependent survival analysis could be considered instead of the Cox model, although stratified analyses may solve the issue in some degree.

2) Competing risk. As the outcomes of the Cox model are attempted suicide and completed suicide rather than all-cause mortality, there exists a competing risk from death on these suicide outcome. Can authors show the percentages of death (all cause mortality) in these cohorts? Fine and Gray competing risk regression model could be considered in this situation.

3) The finding of slightly decreased risk of parental suicidal attempt more than seven years after a child's cancer diagnosis (HR, 0.86; 95%CI: 0.75-0.98) is difficult to comprehend. Apart from all the explanations in the discussion, the authors need to make sure the statistical analyses are robust, especially with the time-dependent and competing risk analyses.

Reviewer #2: This manuscript reports an investigation of the association between childhood cancer and suicide mortality/attempt in parents using data from two countries. My comments are as follows:

1. The sibling comparison obviously has some limitations, in that there's presumably a reasonably high level of shared direct exposure (people going through the trauma of a niece or nephew with cancer) which isn't usually what's wanted in this technique. It could therefore represent 'over-adjustment' and result in overly conservative estimates. Although this doesn't explain the observed persistence of the findings of interest, I think this point is worth making in the Discussion. 

2. I think the Discussion ought to acknowledge the limitation of the cancer aggressiveness measure, since this was aggregated by type of cancer, and didn't take into account individual prognostic markers such as stage at presentation. 

3. It would be helpful to have some information in the Discussion on the likely comprehensiveness of outcome capture from the diagnostic codes - presumably other research has investigated register data for this (i.e., coding accuracy for 'suicide attempt')

4. It rather looks as if '7 years since diagnosis' as an outcome definition was applied following visual inspection of the incidence pattern rather than a priori, so I think this needs an appropriate note of caution, particularly when it's portrayed as the main finding in the abstract. This element of the analysis felt quite exploratory and it didn't help that it was allowed to vary between population and sibling comparisons. 

5. I feel that the lack of suicide mortality difference should be in the abstract, as this was one of the primary comparisons. However, some consideration would be helpful in the Discussion on the statistical power for this (i.e., what hazard ratio (for example, over the initial 7 years for consistency) would have been detectable from the number of events and sample size). 

6. The presence or not of other offspring would seem to be an obvious factor influencing risk of attempted suicide in the parent. Was there no possibility of ascertaining sibling numbers for index children as a covariate?

7. On page 6 - I don't think 'a cancer child' is appropriate terminology. 

Reviewer #3: This manuscript details the use of two large, population-based cohorts to examine the risk of suicidal behavior among parents of children with a cancer diagnosis. This is an important topic of study given the psychological toll on any caregiver of individuals with medical conditions, and in particular parents. Although the study is of course limited to observational data, the authors employ two methods to try to facilitate causal inference: a matching approach and a comparison to the parents' siblings (the latter to address the possibility that some familial factors could jointly increase risk of cancer [passed down to the child] and suicidality [expressed by the parent]). The large sample sizes generally provide statistical power that is unavailable in other studies, and the use of national registry data ensures reasonably consistent, objective, clinically-relevant measures. Another strength of the study is that the authors are able to adjust for potentially important confounders, such as parental psychiatric history. My primary concern is that the authors provide little justification for some of the decisions they made with regard to coding, which leads to questions about whether results are cherry-picked. For example, no explanation is given for splitting the child's cancer into three categories (CNS, hematological malignancy, and other), or for binning parental age into <40, 41-60, 60+. Were these categories pre-specified, or determined only after conducting preliminary analyses (which could be a problem, though could also be justified)? Additional comments are noted below.

1) Please change "completed suicide" to "suicide" or "suicide death" throughout (see https://www.camh.ca/-/media/files/words-matter-suicide-language-guide.pdf and https://psychology.org.au/publications/inpsych/2013/february/beaton).

2) When describing exclusion criteria, I recommend coming up with a different way to describe parents who had a child receive a cancer diagnosis prior to the onset of observation: "cancer child" is not ideal.

3) Further justification for including "deliberate self-harm" in identifying suicide attempt cases is necessary. That terminology typically refers to non-suicidal self-injury, which differs etiologically and epidemiologically from suicide attempt.

4) Some covariates are time-varying (e.g., household income). Were these modeled as time-varying covariates, or was status at beginning of the observation period used? If the latter, were time-dependent coefficients used?

5) What was the rationale for how "cancer type" was split into three groups (CNS cancer, hematological malignancy, other)?

6) The authors state that they observed increased risk of parental suicide attempt in the first 7 years after the offspring's cancer registration, and decreased risk thereafter. Based on the figure, this seems a bit misleading, as confidence intervals appear to always include 1 (the null). This is inconsistent with results reported in Table 2, which indicate that CIs did not include 1 when the HRs were based on the first 7 years versus thereafter. Perhaps this is merely an issue with how the splines were placed for the figure, but more explanation is warranted for the sake of transparency.

7) The authors might consider moving discussion of the results around suicide death to the Supplement; despite the large sample size, suicide death is sufficiently rare that even this study might be underpowered to detect effects.

8) Little note is made of the relatively small effect sizes observed and the degree to which HRs are attenuated in the sibling analysis. This seems to warrant additional discussion/explanation, particularly for readers who are less familiar with how studies including a family design can be used to account for some confounders.

9) The Discussion could be improved if the authors provide some ideas of the implications of their findings. For example, should parents of children <18 who receive a cancer diagnosis be provided with referrals to counselors/therapists who can help them deal with the psychological stress that attends their situation? Given the audience of PLoS Medicine, I imagine that clinical recommendations would be broadly appreciated. Furthermore, this addition would speak to the journal's guideline that papers have "clear implications for patient care, public policy, or clinical research agendas."

[LINK]

---

## [Decision Letter · Decision Letter 2]

10 Nov 2023

Dear Dr. Liu,

Thank you very much for re-submitting your manuscript "Suicide attempt and death by suicide among parents of young individuals with cancer: a population-based study in Denmark and Sweden" (PMEDICINE-D-23-00808R2) for review by PLOS Medicine.

I have discussed the paper with my colleagues and the academic editor and it was also seen again by all 3 reviewers. I am pleased to say that provided the remaining editorial and production issues are dealt with we are planning to accept the paper for publication in the journal.

[LINK]

We look forward to receiving the revised manuscript by Nov 17 2023 11:59PM.   

Best wishes,

Pippa

Philippa Dodd, MBBS MRCP PhD

PLOS Medicine

plosmedicine.org

pdodd@:plos.org

Requests from Editors:

GENERAL

Thank you for your detailed and considered responses to previous editor and reviewer comments. Please see below for further revisions required prior to publication.

* We agree with the reviewer, please see below, that the claim of 7-years as being a critical end point is overstated in places, please temper these claims throughout (in particular the abstract and when reporting your results).

** Please provide further clarification as per the reviewer request (below) regarding parental age classifications. 

REFERENCES

For in-text reference callouts please format as follows, for example, line 74, ‘…and financial challenges [2,3].’ Please note an additional space preceding the opening bracket and the absence of a space between the different citations. Please check and amend throughout.

Line 302 – ‘…95% CI [1.002, 2.10];…’ any reason for reporting as such?

SOCIAL MEDIA

To help us extend the reach of your research, please detail any X (formerly Twitter) handles you wish to be included when we tweet this paper (including your own, your coauthors’, your institution, funder, or lab) in the manuscript submission form when you re-submit the manuscript.

Comments from Reviewers:

Reviewer #1: Thanks authors for their great effort to improve the manuscript. I am satisfied with the response and revision. No further issues needing attention.

Reviewer #2: My comments have been satisfactorily addressed. 

Reviewer #3: The authors have made considerable revisions to the text, which overall result in a manuscript with improved clarity. However, some substantial concerns remain.

1) Despite feedback from the reviewers of the original submission, there remain references in the text to the seven-year mark as a critical end point to the elevated risk of suicide attempt. In one response to reviewers, the authors propose that statistical power is an issue that leads to confidence intervals overlapping 1. This could very well be the case, but if the HR estimate does not differ from 1, then it is not appropriate to suggest that there is an effect. Based on Figure 1, it appears that the CIs overlap 1 well short of seven years beyond the offspring's cancer registration. There is no reason to exaggerate the length of the observed associations, and references to the seven-year mark still need to be scaled back. 

2) The explanation for how parental ages were binned does not make sense to me. If the median age of the parents at cohort entry was 56, how does the middle bin of 41-60 line up with that?

[LINK]

---

## [Editor Report · Decision Letter 3]

14 Nov 2023

Dear Dr Liu, 

On behalf of my colleagues and the Academic Editor, Professor Charlotte Hanlon, I am pleased to inform you that we have agreed to publish your manuscript "Suicide attempt and death by suicide among parents of young individuals with cancer: a population-based study in Denmark and Sweden" (PMEDICINE-D-23-00808R3) in PLOS Medicine.

PRESS

Thank you again for submitting to PLOS Medicine. It has been a pleasure handling your manuscript, we look forward to publishing your paper. 

Best wishes,

Pippa 

Philippa Dodd, MBBS MRCP PhD 

PLOS Medicine